

# Springtime aerosol load as observed from ground-based and airborne lidars over Northern Norway

Patrick Chazette[1], Jean-Christophe Raut[2], and Julien Totems[1]

[1]Laboratoire des Sciences du Climat et de l'Environnement (LSCE), Laboratoire mixte CEA-CNRS-UVSQ, UMR 1572, CEA Saclay, 91191 Gif-sur-Yvette, France
[2]LATMOS/IPSL, Sorbonne Université, CNRS, UVSQ, Paris, France

*Correspondence to*: Patrick Chazette (patrick.chazette@lsce.ipsl.fr)

**Abstract.** To investigate the origin of springtime aerosols in the Arctic region we performed ground-based and airborne 355 nm-Raman lidar observations in the North of Norway (Hammerfest). Two lidars were embedded (i) on an ultra-light aircraft for vertical (nadir) or horizontal line-of-sight measurements, (ii) in an air-conditioned van on the ground for vertical (zenith) measurements. This field experiment was designed as part of the Pollution in the ARCtic System (PARCS) project of the French Arctic initiative, and took place from 13 to 26 May, 2016. The consistency between lidar measurements is verified by comparing nadir, horizontal line-of-sight, and ground-based Raman lidar profiles. Dispersion of the order of 0.01 km⁻¹ is obtained between lidar-derived aerosol extinction coefficients at 355 nm. The aerosol load measured in the three first kilometers of the troposphere remains low throughout the campaign, with aerosol optical thickness (AOT) $\lesssim$ 0.1 at 355 nm (~0.05 at 550 nm). The main contributors to the evolution of the aerosol load at low altitude prove to be one of the flares of the nearby Melkoya gas processing facility, the oceanic source and the transport of aerosols from industrial sites in Russia. Moreover, ground-based lidar measurements allowed the possibility to identify three cases of long-range aerosol transport (between 3 and 8 km above the mean sea level). Using back trajectories computed with the Lagrangian model FLEXPART-WRF, these aerosol plumes are shown to be the result of the strong forest fires that occurred in the area of Fort McMurray, in Canada. They can, at most double the AOT value over the Arctic area, with an anomaly of 0.1 on the AOT at 355 nm.

**Keywords:** PARCS, Raman, lidar, ULA, airborne, aerosol, optical properties, back trajectories

## 1    Introduction

The pristine Arctic environment is very sensitive and can be easily disturbed by anthropogenic activities, with irreparable consequences. Anthropogenic aerosols play a major role in the evolution of the Arctic radiative balance, as pointed out by the IPCC (IPCC, 2014), and have to be better quantified. Moreover, the Arctic region is exposed to thin but persistent haze (Breider et al., 2014; Shaw, 1995), as well as episodic events of carbonaceous aerosol plumes in the free troposphere (Brock et al., 2011; Quinn et al., 2008; Warneke et al., 2010) since the industrial era. This environmental challenge posed by tropospheric aerosols in the Arctic has already been pointed out by Barrie (1986) and, even more recently by authors as Law et al. (2017) or Yang et al. (2014), who analyzed the climatic impact and showed that aerosols induce a warning of about 0.6 K decade⁻¹.

Following these observations, the French Arctic initiative project Pollution in the ARCtic System (PARCS) was performed to improve our understanding of aerosols in the Arctic troposphere. A point of focus was the long-range



transport of anthropogenic and biomass burning aerosols over the Arctic region. This innovative field campaign
took place from 13 to 26 May, 2016 in the region of Hammerfest (70°39′45″N 23°41′00″E, Norway), 90 km
southwest of the North Cape, within the Arctic Circle. It involved ground-based and airborne Raman lidar
observations. The mesoscale dynamic modeling was performed using the Weather Research and Forecasting
(WRF) model (Skamarock et al., 2008).
The PARCS experiment follows several international initiatives such as the recent Arctic Climate Change,
Economy and Society (ACCESS) over Northern Norway in July 2012 (Raut et al., 2017). ACCESS itself followed
the international Polar Study initiatives using Aircraft, Remote Sensing, Surface Measurements and Models,
Climate, Chemistry, Aerosols and Transport (POLARCAT) in 2008 (Ancellet et al., 2014), and the Arctic Research
of the Composition of the Troposphere from Aircraft and Satellites (ARCTAS) in 2008 (Jacob et al., 2009).
Obviously, the PARCS experiment is a snapshot of the aerosol situation in Northern Norway. As in all field
campaigns, the atmospheric environment is sampled over a short period of time and is not necessarily
representative of the local and seasonal meteorological conditions. The PARCS experiment took place during
large-scale weather conditions disturbed by the strong El Niño of 2015-2016 (Hu and Fedorov, 2017), which led
to temperatures in the Arctic planetary boundary layer (PBL) 3 to 4 °C above the 10-year normal climatic
conditions. Also associated with such exceptional atmospheric conditions, transport in the high troposphere
favored the presence of air masses from North America. Spring 2016 was marked by extreme wildfires in Canada's
Alberta territory, close to Fort McMurray (Kochtubajda et al., 2017; Landis et al., 2018). The coupling between
pyro-convection (Fromm et al., 2005; Peterson et al., 2015) and large-scale atmospheric transport may inject large
quantities of aerosols into the upper troposphere (Ancellet et al., 2016), whose lifetime greatly exceeds a week in
the absence of precipitation throughout their transport. Part of these aerosol layers were sampled by a ground-
based Raman lidar, which made it possible to describe both the vertical structure and the optical properties of the
aerosol plumes (Chazette et al., 2014), but also the history of their transport using the synergy between the Cloud-
Aerosol LIdar with Orthogonal Polarization (CALIOP) (Winker et al., 2003), the Moderate Resolution Imaging
Spectroradiometer (MODIS) (King et al., 1992) spaceborne instruments, and mesoscale modeling. The observation
of biomass fire aerosol transported at high altitude over long distances has already been reported by several authors
for different regions of the Earth (Ancellet et al., 2016; Formenti et al., 2002; Forster et al., 2001; Paris et al., 2009;
Quennehen et al., 2011; Sitnov and Mokhov, 2017). During the POLARCAT summer campaign in 2008, (Schmale
et al., 2011) and (Thomas et al., 2013) characterized aerosol and gas pollution from fire plumes transported from
North America to Greenland.  Franklin et al. (2014) and Taylor et al. (2014) documented a case study of aerosol
removal in a biomass burning plume over eastern Canada in 2011. More recently, the long-range transport of
aerosols from Siberia has also already been evidenced (Marelle et al., 2015; Sitnov and Mokhov, 2017). During
the ACCESS airborne campaign in summer 2012 (Roiger et al., 2015), extensive boreal forest fires resulted in
significant aerosol transport to the Arctic (Raut et al., 2017). These plumes originating from Siberian wildfires are
very common during late spring and summer, and they may be mixed with aerosols coming from highly polluting
industrial sources such as oil and gas rigs, or petroleum refineries. Vaughan et al. (2018) describe the transport of
biomass burning aerosols over the United Kingdom originating from extensive and intense forest fires over Canada
in spring 2016. It should be noted that all previous authors only reported isolated long-distance transport events
and that this type of phenomenon is rare; the probability to observe one during the short duration of the PARCS



campaign was low. The chosen period for PARCS associated with a strong El Niño certainly favored long-range
transport of aerosols and offered an opportunity to sample 3 different tropospheric plumes.
This paper focuses on the long-range transported aerosols observed during the PARCS campaign as well as the
evolution of the aerosol load in the low troposphere. The field experiment is presented in Section 2, where ground-
based and airborne measurements are described. The large-scale observations derived from spaceborne
instruments and mesoscale modeling are presented in Section 3. Section 4 is devoted to the description of the
aerosol structures observed during the field campaign, with a spotlight on the low troposphere. Section 5 is
dedicated to the identification of the origins of the high-altitude aerosol plumes. The data coherence is discussed
in Section 6 and the conclusion is presented in Section 7.
**2    Field experiment**
The aerosol load is investigated using observations gathered from 13 to 26 May, 2016, during the PARCS field
campaign held in Northern Norway, over 70°N (Figure 1). The ground-based van MAS (Mobile Atmospheric
Station (Raut and Chazette, 2009)) and an ultra-light aircraft (ULA) were mainly equipped with active remote
sensing instruments (Figure 2): the Weather Atmospheric Lidar (WALI) and the Lidar for Automatic Atmospheric
Survey Using Raman Scattering (LAASURS), respectively.
We selected an experimental site near Hammerfest, next to the airport. The main reason for this is that the Melkoya
gas processing facility, which is the northernmost coastal installation and uses the latest techniques of LNG
(Liquiefied Natural Gas), has two potentially active flares that could significantly influence atmospheric aerosol
concentrations: a high-pressure flare from processing and a low-pressure flare from loading and storing LNG. In
addition, with the local and shipping activities, the region may be subject to the advection of air masses from the
Murmansk area, which has a large concentration of oil and gas industries. We benefited from the help of the Avinor
crew of Hammerfest Airport in order to have a suitable operating base and all the necessary power supply. They
also helped us navigate the ULA, freely lent their hangar on the airport and offered staff support.





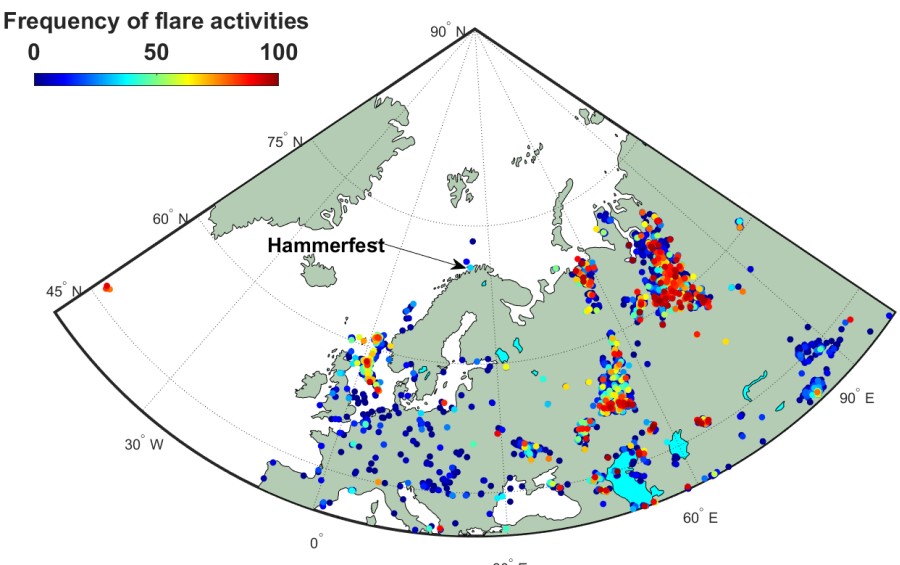


**Figure 1: Location of the ground-based measurement site, close to Hammerfest (Norway). The frequencies of the main**
**flares activities for both oil and gas rigs are given following (Elvidge et al., 2016) for 2016.**

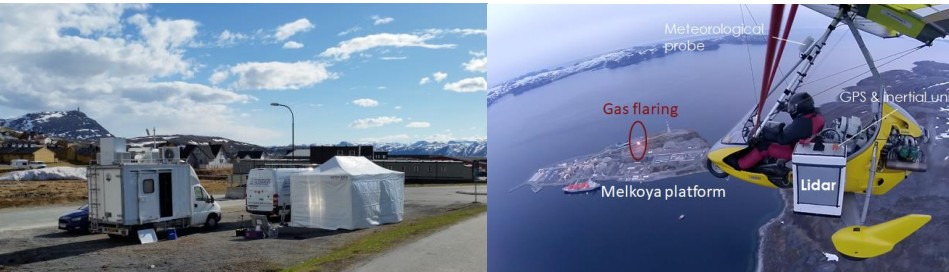


**Figure 2: Left picture: Mobile atmospheric station (MAS) located near the Hammerfest airport, equipped with the**
**WALI Raman lidar. Right picture: N$_2$-Raman lidar LAASURS embedded on a ULA. The ULA is flying over the**
**Melkoya platform where a gas flaring is active.**
**2.1    Ground-based measurements**
Figure 2 shows the MAS, located close to the Hammerfest airport. A schematic representation of the MAS and its
onboard instruments is given in Figure 3. It was equipped with the 354.7 nm water vapor Raman lidar WALI
(Chazette et al., 2014). These instruments carried out continuous measurements from 13 to 26 May, 2016, with a
final vertical resolution of 15 m and 1-minute integration (~1000 laser shots). The main characteristics of WALI
are summarized in Table 1.



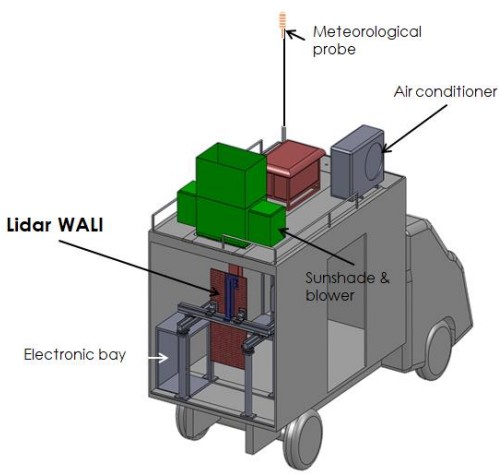

**Figure 3: Schematic representation of the MAS equipped with the Raman lidar WALI.**
**Table 1**: **Raman lidar WALI and LAASURS main characteristics. In the third column the corresponding characteristics**
**of the spaceborne CALIOP lidar are also presented.**

|  | WALI | LAASURS |
|---|---|---|
| Carrier | Ground-based (truck) | Airborne |
| Laser | Nd:YAG, flash-pumped, Q-switched  Q-smart QUANTEL | Nd:YAG, flash-pumped, Q-switched  Ultra QUANTEL |
| Pulse length | <10 ns | 6 ns |
| Emitted energy | 120 mJ at 355 nm | 30 mJ at 355 nm |
| Frequency | 20 Hz | |
| Reception channels | // 355 nm  ⊥ 355 nm  $N_2$-Raman 387 nm  $H_2O$-Raman 407 nm | // 355 nm  ⊥ 355 nm  $N_2$-Raman 387 nm |
| Reception diameter | 15 cm | |
| Field-of-view | ~2.3 mrad | |
| Full overlap | ~200 m | |
| Filter bandwidth | 0.2 nm | |
| Detector | Photomultiplier tubes | |
| Post processing vertical resolution | 15-30 m | |

**2.2    Airborne measurements**
In order to sample the low troposphere around the ground-based lidar, the ULA/Tanarg-embedded Raman lidar
system LAASURS was used (Chazette and Totems, 2017). Lidar containment enabled operation for temperatures





down to ~ -17 ° C, but with a loss of nearly 40% of the emitted energy. This has greatly limited the altitude
explorations above 1 km above the mean sea level (AMSL) and we have essentially worked just above the PBL.
The lidar and the ULA's flights close to the Melkoya platform are represented in Figure 4.
The aircraft, Tanarg 912 XS, was built by the Air Création Company (http://www.aircreation.fr/) and offers a
maximum total payload of ~250 kg (Table 2). Flight durations were between 1 and 2 hours, depending on flight
conditions, with a cruise speed around 85-90 km h$^{-1}$. The ULA is also equipped with i) a VAISALA 300
meteorological probe for temperature, pressure and relative humidity, ii) a Global Positioning System (GPS) and
an Attitude and Heading Reference System (AHRS), which are part of the MTi-G components by XSens. The
lidar, whose characteristics are given in Table 1, is designed to fulfill eye-safety standards (EN 60825-1). The wide
field-of-view (FOV) ~2.3 mrad allows a 90% overlap of the transmission and reception paths beyond ~ 200 m
with the desired setting for the experiment. After correction of the overlap function, the data can be used from 150
m with a negligible error compared with the one due to signal noise. The acquisition was performed by averaging
400 laser shots leading to a temporal sampling close to 25 s.
**Table 2: Tanarg 912 XS ULA main flight characteristics.**

| ULA flight characteristics | |
|---|---|
| True airspeed: 17 to 40 m s$^{-1}$ (60 to 145 km h$^{-1}$) | Endurance: 3 hr (max 4 hr at 20 m s$^{-1}$) |
| Ascent speed: up to 365 ft min$^{-1}$ (110 m min$^{-1}$) | Maximum scientific payload: 120 kg |
| Descent speed: 825 ft/min (250 m min$^{-1}$) | Maximum altitude: 5.8 km |

**2.3   Strategy and flight plans**
We performed a total of 14 flights during the field campaign. The majority of flights were performed near the
airport, around the Hammerfest peninsula. Four flights were particularly interesting for aerosol layers detection
(Table 3). Three flights were not successful because of technical difficulties and the other ones were performed in
low-cloud conditions, with condensation at the ceiling altitude. Only one day out of 3 was not very cloudy over
the period of measurements. The more exploitable flights were performed during nighttime. Note that during the
field campaign, the sun did not go down under the horizon. Each flight included a slow spiral ascent or descent
where the lidar was aiming horizontally, and once at the ceiling altitude, the lidar was rotated to aim at the nadir.
Flight 4 passed very close to the Melkoya platform and permitted the sampling of one active flare. Flights 10 and
11 were around the Hammerfest peninsula for 2 non-consecutive hours to check the representativeness of the site
for aerosols trapped within the PBL. For flight 13, the ULA took-off from Hammerfest airport at 21:38 UTC
(universal time count) and headed towards North-Cape at the ceiling altitude of ~1.8 km AMSL. Before reaching
North-Cape, the ULA changed heading and flew parallel to the coastline before veering towards the airport, where
it landed at 23:58 UTC.
**Table 3: Flights information: identification, date and description.**

| Flight identification | Date & hour (UTC) | Description |
|---|---|---|
| 4 | 16 May, 22:39-23:24 | Flight along the west coast of the Hammerfest peninsula overflying the Melkoya platform in cloudy condition. |





| | | |
|---|---|---|
| 10 | 20 May, 18:56-20:00 | Flight around the Hammerfest peninsula in cloud free condition. |
| 11 | 20 May, 23:02-<br>21 May, 00:26 | Flight around the Hammerfest peninsula in cloud free condition. |
| 13 | 22 May, 21:38-23:58 | Flight towards North-Cap in cloud free condition. |


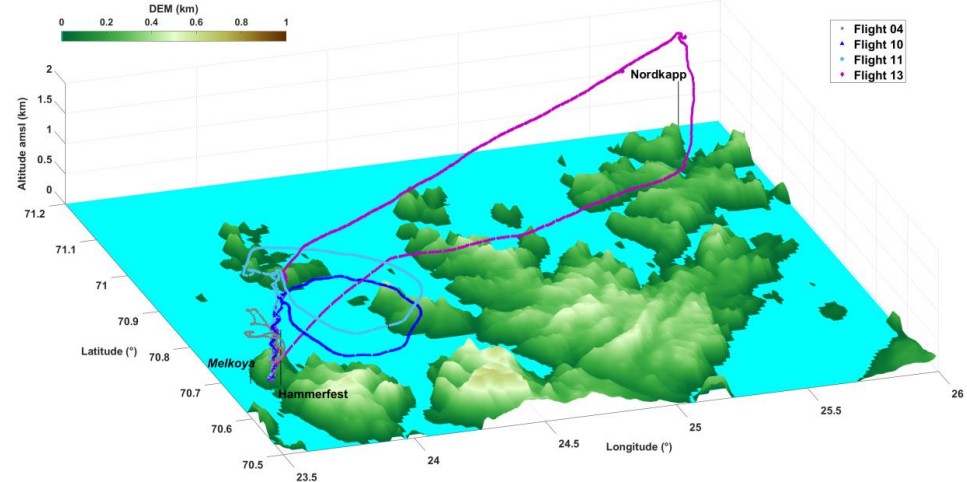


**Figure 4: Flight plans used for this study: flight 04 on 16 May, flights 10 and 11 on 20-21 May, and flight 13 on 22 May (see Table 3). The flight plans are drawn over the 30 arc-second digital elevation model (DEM) GTOPO30 (https://lta.cr.usgs.gov/GTOPO30).**

### 2.4 Data processing for lidar measurements

Lidar data analyses are not presented in detail hereafter, since the methods used have already been published (e.g. Chazette et al., 2015, and references therein). The aerosol extinction coefficient (AEC), the backscatter to extinction ratio (BER, inverse of the lidar ratio (LR)) and the particle depolarization ratio (PDR) are derived following Chazette et al. (2014) and references therein. The absolute uncertainties on the AEC are ~0.01 km$^{-1}$ and the ones on the PDR are ~1-2% for AEC > 0.03 km$^{-1}$. The absolute uncertainty on the BER (LR) is ~0.004 sr$^{-1}$ (~10 sr) for a mean BER (LR) of 0.020 sr$^{-1}$ (50 sr). It decreases when the BER decreases.

The inversion of nadir lidar profiles acquired from the ULA is more difficult due to the noise level. For this reason, we have limited altitude excursions between 1 and 2 km AMSL. The horizontal measurements of the elastic channel are inverted to retrieve the AEC within an absolute uncertainty of 0.01 km$^{-1}$ following Chazette and Totems (2017) and references therein. We consider a distance from the ULA between ~0.3 and 1.5 km after correction of the overlap function for the calculations. The nadir measurements are inverted using the constraint brought by the horizontal laser shots and the BER derived from the ground-based lidar. We therefore assume that



the aerosol typing does not change during the flight. Note that the N₂-Raman channel of the airborne lidar is too
noisy to be relevant, mainly due to the loss of emitted energy in low ambient temperature.
**3    Large-scale data**
**3.1    Spaceborne observations**
Active and passive spaceborne measurements were used to follow the aerosol plume transport. The horizontal
dispersion of the aerosol plume and its progression along the transport are highlighted with Moderate Resolution
Imaging Spectroradiometer (MODIS, (King et al., 1992; Salmonson et al., 1989)) onboard the polar-orbiting
platforms Terra and Aqua. We used a combination of the aerosol optical thickness (AOT) at 550 nm derived from
the two satellites. The level 2 products are provided with a spatial horizontal resolution of 10×10 km²
(http://modis.gsfc.nasa.gov). The uncertainty on the AOT is *±0.15±0.05 AOT* over land and *±0.05±0.03 AOT* over
ocean (Chu et al., 2002). The vertical structures of the aerosol layers over their sources are derived from Cloud-
Aerosol LIdar with Orthogonal Polarization (CALIOP) aboard Cloud-Aerosol Lidar and Infrared Pathfinder
Satellite Observations (CALIPSO, http://www-calipso.larc.nasa.gov, (Winker et al., 2007)). We have used the
4.10 version of CALIOP level-2 data. We mainly took into consideration the aerosol typing of (Burton et al.,

182    2015).

**3.2    Modeling strategy**
*3.2.1    Weather model*
The 3.5.1 version of the regional non-hydrostatic Weather Research and Forecasting (WRF) model (Skamarock et
al., 2008) has been used for weather simulations along the field campaign. The model was run from 7 May, to 28
May, 2016, with a dynamical time step of 3 min on a polar stereographic grid almost encompassing the Northern
Hemisphere (> 7°N). The domain has 300x300 grid points with a horizontal resolution of 50 km and 50 vertical
levels up to 50 hPa, considered as the top-of-atmosphere pressure. The initial and boundary meteorological
conditions for this hemispheric domain are provided by the 6-hourly operational analyses of the ECMWF/IFS
NWP model (Dee et al., 2011) from the European Centre for Medium-range Weather Forecasts (ECMWF), with
the support of the ESPRI (Ensemble de Services Pour la Recherche à l'IPSL, https://www.ipsl.fr/Organisation/Les-
structures-federatives /ESPRI) team. Nudging has been applied above the planetary boundary layer (PBL) to wind,
temperature and humidity fields, with an update time of 6 hours. The parameterizations used are described in (Raut
et al., 2017) and (Marelle et al., 2017). Briefly, the prognostic turbulent kinetic energy scheme of Mellor-Yamada-
Janjic (MYJ) is used for the boundary layer, with the associated Janjic Eta surface layer module (Janjić, 1994).
Land surface processes are resolved using the Noah LSM (unified Noah land surface model (Chen and Dudhia,
2001)). We have used the Morrison 2-moment scheme (Morrison et al., 2009) to calculate cloud microphysical
properties and grid-scale precipitation. Subgrid clouds are represented using the Kain-Fritsch with Cumulus
Potential parameterization developed by (Berg et al., 2013). The shortwave and longwave radiation calculations
are performed using the RRTMG scheme (Rapid Radiative Transfer Model for Global applications; (Iacono et al.,

202    2008)).



### 3.2.2  Back-trajectories

The Lagrangian particle dispersion model FLEXPART-WRF (Brioude et al., 2013) derived from the FLEXPART model (Stohl et al., 2005) is run in this study to investigate the origin and transport pathways of air masses bringing aerosols to Hammerfest. Three backward simulations are performed on 15 May, 05:00 UTC, 20 May, 20:00 UTC and 22 May, 21:00 UTC to provide insight into the representation of aerosol transport to Scandinavia. In each of them, a total of 10 000 particles are released at Hammerfest in a volume of 50 km x 50 km large and 1 km (200 m) thick for 15, 20 May (22 May) centered on the aerosol plumes detected aloft. The origin of each air parcels is then established using the meteorological fields simulated by WRF (Sect. 3.2.1). As transport durations are typically less than 9 days, this approach finally allows us to track the air mass origin over the source regions of interest. As a proxy to represent the source-receptor relationships, we use the PES (potential emission sensitivities) that quantify the amount of time spent by the particles in every grid cell.

## 4  Aerosol observed in the Arctic troposphere

There are few clear sky periods during the campaign, as is often the case over the studied area. The interesting periods are given in terms of AEC and PDR in Figures 5 to 7 (14-15, 20-21, and 22-23 May, 2016), where outstanding high-altitude features are highlighted. The temporal evolutions of the AEC profile are given in local time (LT) corresponding to UTC+2.

### 4.1  Optical properties of aerosol layers derived from the ground-based lidar

The coupling between the elastic and the $N_2$-Raman channels is used to derive the BER for the different aerosol layers. The molecular contribution is corrected using the hourly vertical profiles of temperature derived from WRF and a classical modeling of the Rayleigh scattering (Bodhaine et al., 1999). The troposphere has been divided into two altitude ranges, as the lower and upper layers are not necessarily composed of the same aerosol types. The first aerosol layer is located between the ground level and ~2.5-3 km AMSL and the second one above 3 km AMSL. The retrieval of the BER for each layer and each measurement period is given in Figure 8. The correct estimate of the BER is obtained when the optical thickness derived from the elastic channel of the lidar is very close to that deduced from the $N_2$-Raman channel (Chazette et al., 2017).

On 14-15 May, the mean BER is ~0.018 $sr^{-1}$ for the upper layer with a standard deviation of 0.002 $sr^{-1}$ (now noted ~0.018±0.002 $sr^{-1}$), whereas as BER is ~0.028±0.003 $sr^{-1}$ in the lower troposphere (Figure 8a). Due to the uncertainty linked to the overlap function, the sensitivity of  the first 200 m where marine aerosols may significantly contribute is lesser. Nevertheless, the higher value observed in the vicinity of the PBL is likely to be associated with a contribution of marine aerosols (BER~0.04 $sr^{-1}$ (Flamant et al., 1998a)). The bottom layer depolarizes very slightly the lidar signal, with PDR <3% and even highlights a lower signature (~1.5%) after 0230 LT. It may be due to a larger oceanic contribution, which leads to an increase of the AEC in the PBL (~0.04 $km^{-1}$). The upper layer has slightly higher PDR values, of the order of 5-6%. Within this range of PDR, the particles cannot be dust-like aerosols. Nonetheless, they are likely to be pollution or biomass burning particles transported toward the measurement site. The total AOT, without the upper layer, is close to 0.08 at 355 nm and increase up to ~0.2 in presence of the higher aerosol plume (Figure 5).





The BER is smaller (0.012±0.002 sr⁻¹, Figure 8b) for the upper layer on 20-21 May, a typical value expected for
pollution and/or biomass burning aerosols. The PDR is also smaller with a mean value close to 1.5%. The aerosols
in the lower troposphere exhibit a larger BER (0.037±0.003 sr⁻¹), demonstrating a strong influence of the oceanic
sources. There are also associated with a small PDR, ~1%. The AOT in the lower atmosphere is similar to the one
on 14-15 May. The elevated aerosol plume presents an excess AOT close to 0.1 at its maximum (Figure 6).
The third period of interest (22-23 May) shows a tiny plume in the middle troposphere, between 3 and 4 km AMSL
(Figure 7), with a very small AOT excess (~0.03). The BER (Figure 8c) and PDR are similar to the ones of 20-21
May, 0.013±0.002 sr⁻¹ and ~2%, respectively. The layer underneath is less influenced by marine aerosol and shows
a BER close to 0.014±0.003 sr⁻¹, more characteristic of polluted particles. Nonetheless, the layer under 400 m
AMSL is more difficult to sample by the lidar and may contain a significant contribution of marine aerosols, as
suggested by the slight decrease in PDR (Figure 7b).

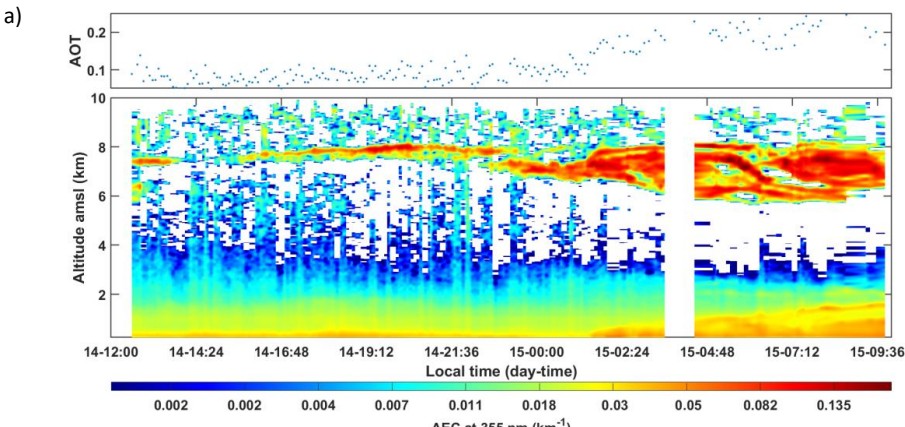

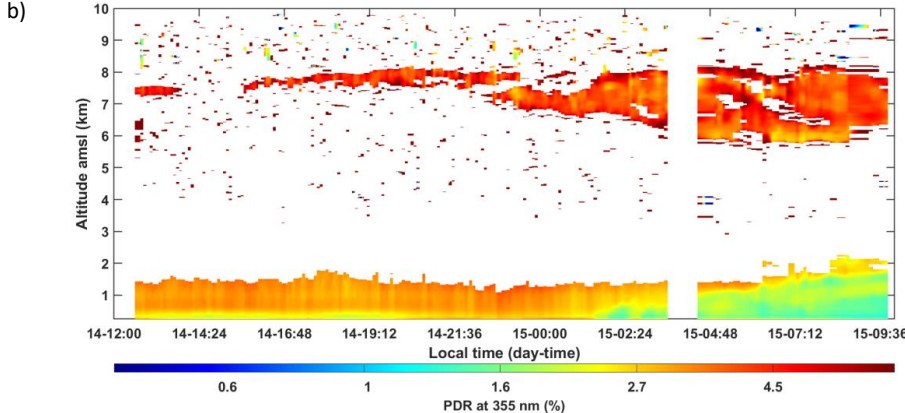

**Figure 5: Temporal evolutions of a) the lidar-derived aerosol extinction coefficient (AEC) and the aerosol optical**
**thickness (AOT); b) the particle depolarization ratio (PDR), at the wavelength of 355 nm, from 14 to 15 May, 2016.**





a)

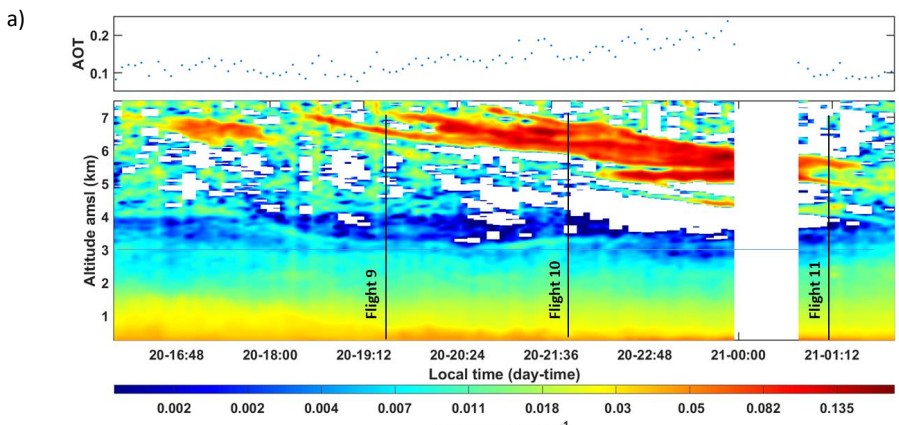

b)

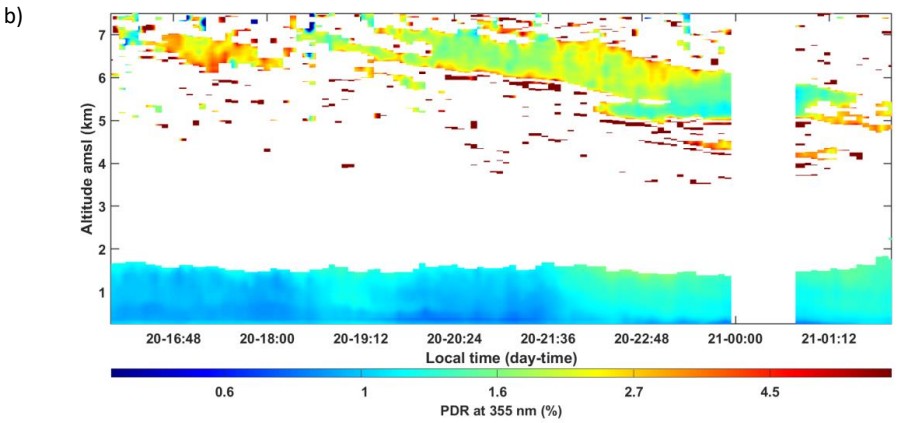

**Figure 6: As Figure 5 but from 20 to 21 May, 2016.**



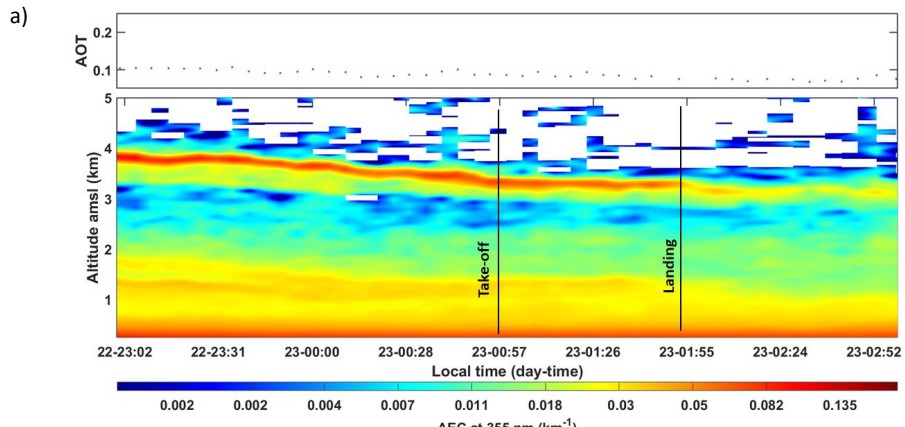

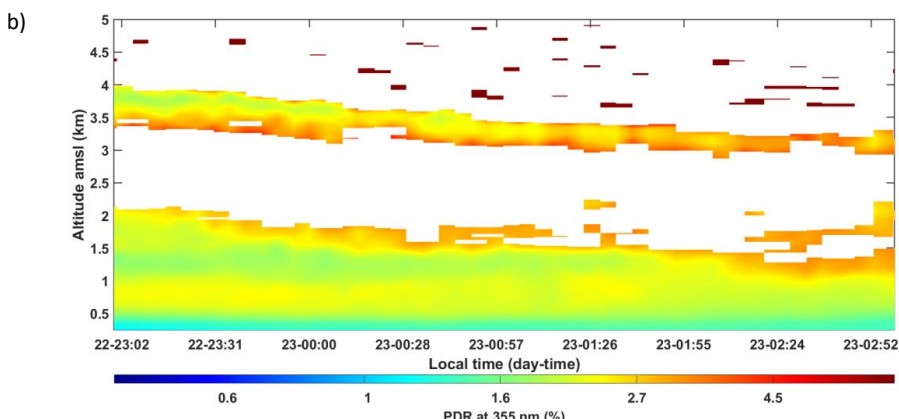

**Figure 7: As Figure 5 but between 22 and 23 May 2016.**





a)

b)

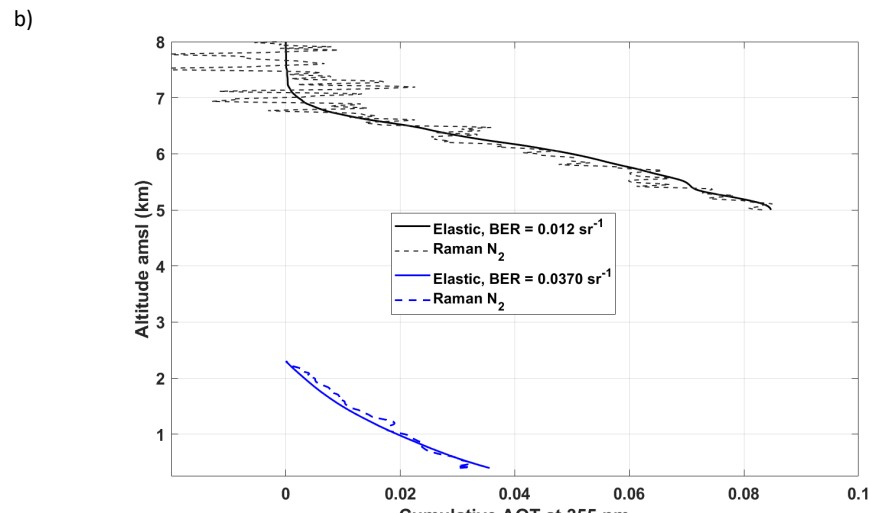



c)

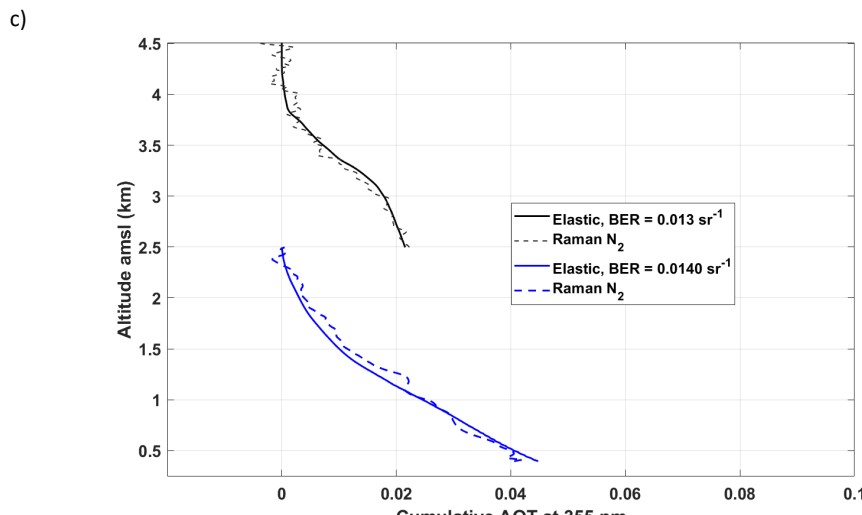

**Figure 8: Cumulative aerosol optical thickness (AOT) derived from both the N$_2$-Raman (dashed line) and the elastic (continuous lines) channels for the upper (black lines) and the lower (blue line) aerosol layers at 355 nm: a) 14-15 May; b) 20-21 May; c) 22-23 May.**

**4.2    Homogeneity of aerosol layers within the lower troposphere**

The lidar-derived aerosol optical properties in the lower troposphere look like homogeneous structures that can be related to the specific situation of the ground-based site. Different sources of aerosols may influence the PBL, the main ones being marine aerosols and anthropogenic aerosols generated in the Hammerfest region (domestic combustion, industrial activity, shipping emissions). To verify the representativeness of the local measurements, we used lidar measurements from the ULA.

*4.2.1    Marine contribution*

The AEC retrieved for flights 10 and 11 are given in Figure 9 with the mean vertical profiles between the ground level and the ceiling flight altitude in both cases, AOTs are low with a small variability of the order of 0.05±0.01. Higher AECs are observed in the northeastern part of the flights (red areas). Because we did not detect many ships in this area, those AEC enhancements are probably due to sea-salts. They may be transported over the nearby coast as the result of the interactions between wind surface and sea (Blanchard and Woodcock, 1980; Flamant et al., 1998b). We note that local pollution is missing altogether.





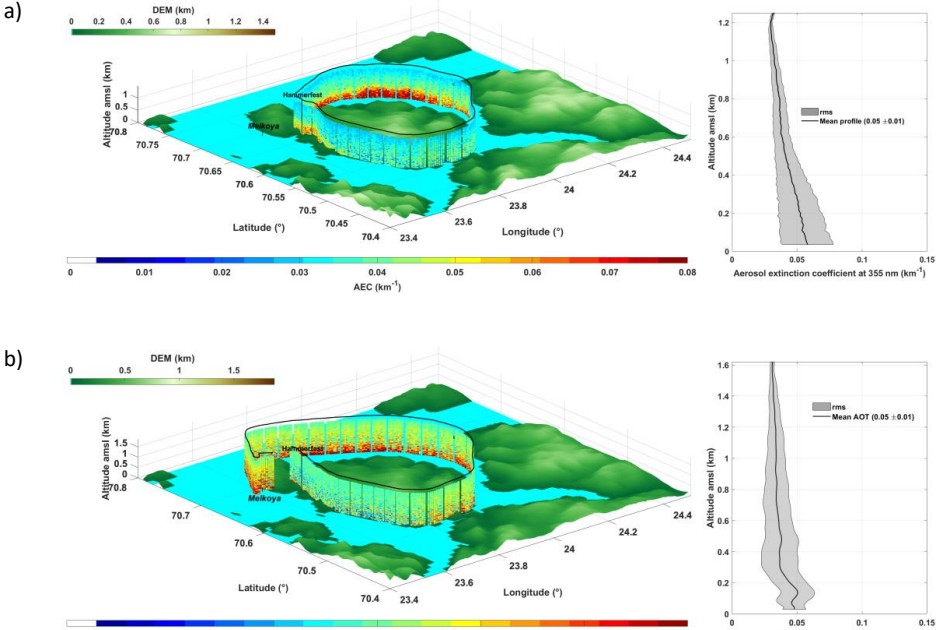

**Figure 9: Vertical profiles of the aerosol extinction coefficient (AEC) derived from the lidar onboard the ULA: a) flight 10 and b) flight 11. The mean AEC vertical profiles and their dispersions are given on the right table. As in Figure 4, the flights are plotted over the digital elevation model (DEM) GTOPO30.**

### 4.2.2 *Gas flaring contribution*

The proximity of the gas rig from the Melkoya facility suggests the presence of an industrial source of aerosol and needed to be quantified. The lowest chimney (~46 m, 70°41'20" N 23°35'59" E) of the Melkoya site used for the low-pressure flare was regularly active during the field experiment and more especially on 16 May (flight 4). The flare (Figure 10, right picture) at the time of sampling was ~20 m above the chimney, with a width ~5 m. On that day, flare smoke presented some blackish color because some hard hydrocarbons (condensate) were present in flare gas. The flight pattern shown in Figure 4 is elongated in Figure 10 using profile number for the sake of clarity. The locations of the ULA when it was close to the flare are highlighted (profiles ~#18 and ~#154) and correspond to the higher AEC of ~0.07 km$^{-1}$. For the second pass, the flare plume is detected from its emission source. The contribution of this flare emission to the AOT is low, ~0.02 at 355 nm for a total AOT between the ground level and 1 km AMSL of ~0.04. The calculation has been done with a BER ~0.037 sr$^{-1}$ and may be underestimated by a factor of 2, as experimental means for a better constraint do not exist. Nevertheless, we note that taken individually, it is a small contribution to the local pollution (representing half of the aerosol background in the first kilometer) and it is very localized in space.



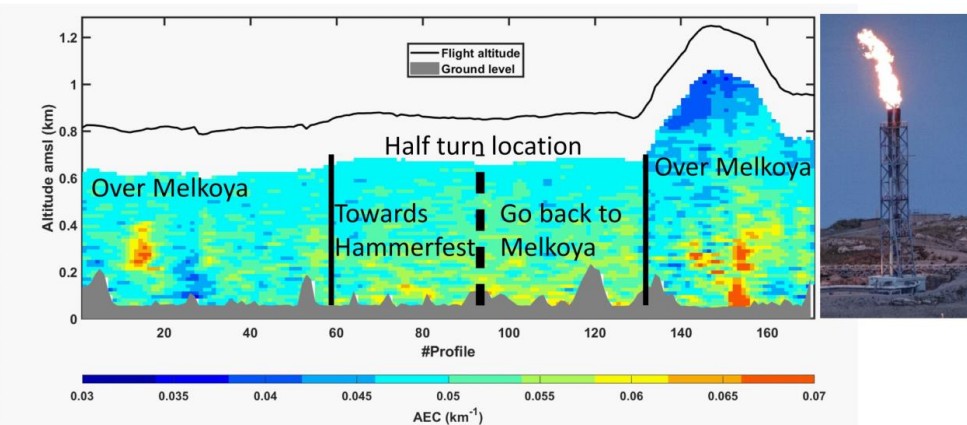

**Figure 10: Left table: Vertical profile of the aerosol extinction coefficient (AEC) during the flight 4 (16 May, 2239-2324**
**local time) dedicated to the sampling of the Melkoya flare at 355 nm. Right picture: Flare sampled by the airborne lidar**
**over the Melkoya platform.**
### *4.2.3    Northern contribution*
During the duration of the experiment, we did not observe any specific contribution to the aerosol load in the
lowest troposphere above the PBL. An exception was for Flight 13 on 22 May, 21:38-23:58 UTC, which was the
longest flight we performed. The vertical profiles of the derived AEC following this flight are plotted on Figure
11a. In the first part of the flight, we note an increase in the AEC close to the ceiling altitude of ~1.7 km AMSL
with values over 0.07 km$^{-1}$. Similar values are measured throughout the flight above the PBL (in red in Figure
11a). The AOT is ~0.06 above the continent and decreases above the ocean (~0.04). The means of constraint are
also limited in this case, because the signal-to-noise ratio for the $N_2$-Raman channel was not high enough and a
BER of 0.014 sr$^{-1}$, initially derived from the ground-based lidar, has been used. The measurements performed
during the flight whilst aiming horizontally are also used as constraints. The aerosol layer has been identified as
coming from the Murmansk region, Russia. The air mass moves along the coast from east to west, drawn by a low
off the Norway coast along the Greenwich meridian. This low is clearly visible in the Figure 11b and is responsible
for the air mass curvature before its northward motion towards Hammerfest and the North Cape.





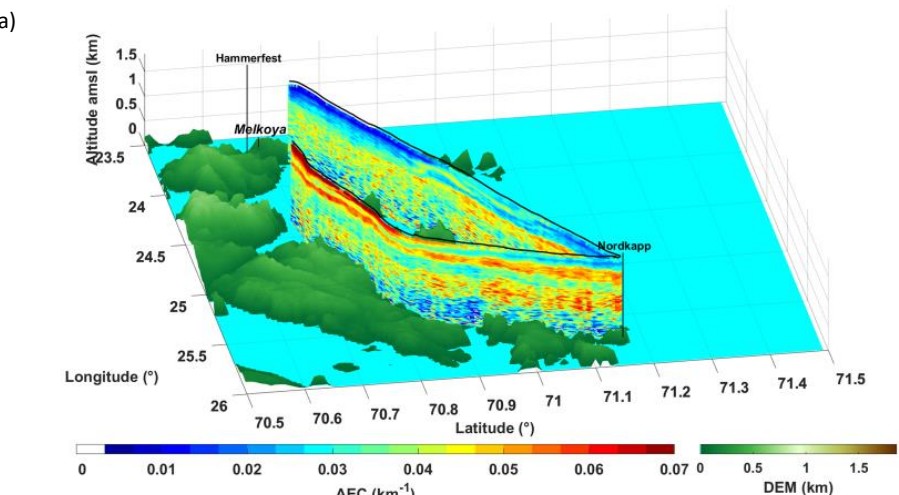

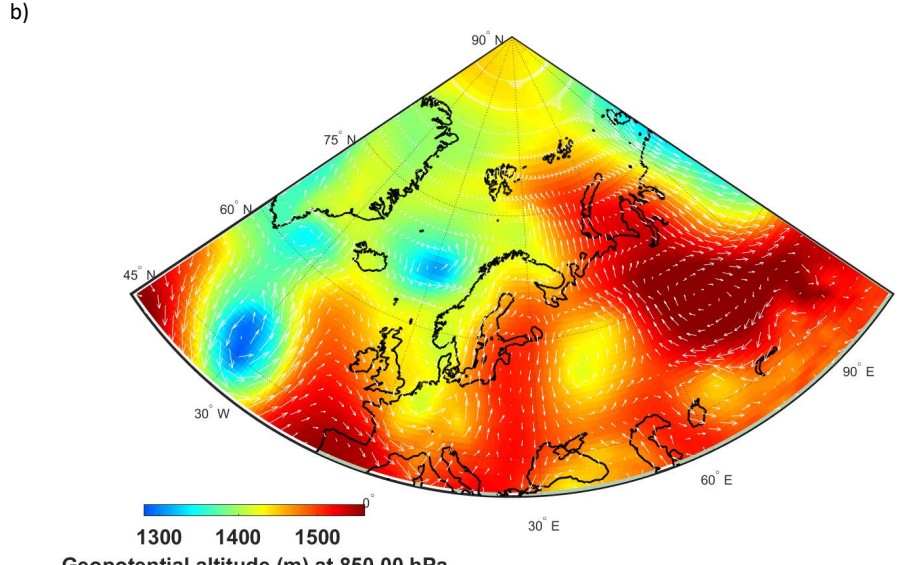

**Figure 11: a) Vertical profiles of the aerosol extinction coefficient (AEC) derived from the lidar onboard the ULA for flight 13 on 22 May, 21:38-23:58 UTC. As in Figure 4, the flights are plotted over the digital elevation model (DEM) GTOPO30. b) Geopotential altitude for the pressure level of 850 hPa (~1.6 km AMSL). The wind field at 850 hPa is also indicated in white arrows.**

## 5 Origin of the upper tropospheric aerosol plumes

To investigate the origin of the three upper aerosol layers, 9-days back trajectories have been performed using

FLEXPART-WRF and constrained by the meteorological fields simulated by WRF over the Arctic region. The





results are given in terms of PES in Figure 12. These simulations are compared, where possible, with the MODIS

and CALIOP space observations to confirm the result.

a)

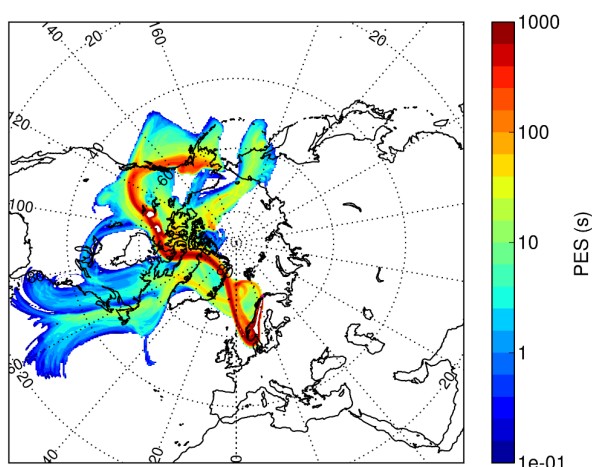

b)

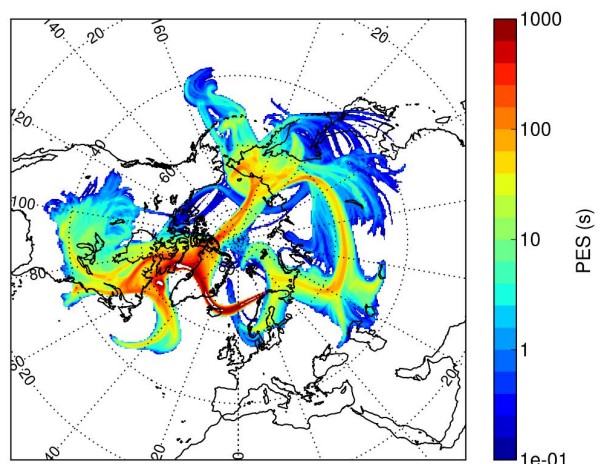





c)

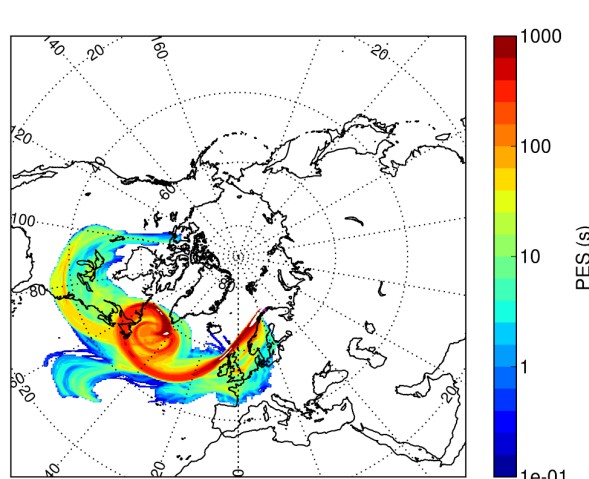

**Figure 12: 9-days back trajectories for the upper aerosol plume observed over Hammerfest on: a) 14-15 May, b) 20-21 May and c) 22-23 May, 2016. The back trajectories are given in terms of potential emission sensitivity (PES).**

### 5.1    Aerosol plume on 14-15 May

On May, 8-9, an aerosol plume was injected in the higher troposphere following the strong forest fires which occurred close to Fort McMurray (56.72°N 111.38°W, North-Eastern Alberta, Canada). As shown in Figure 13, the aerosol plume has been sampled by MODIS on 8 May, with an AOT larger than 0.4 at 550 nm. In the same figure, the thermal anomalies derived from MODIS are also given for both the nominal and the high confidence levels. The aerosol typing derived from CALIOP is plotted in Figure 14a. It confirms the injection of biomass burning aerosols between 6 and 7 km AMSL. The plume then moves north-west of Hudson Bay and reaches Baffin Sea on 12 May. It then crosses Northern Greenland and goes on to cross the Greenland Sea on 13 May. A pronounced northerly flow finally brings the plume to Hammerfest, bypassing the low pressure system located off Norway and responsible for the plume curvature. Elevated smoke aerosols are identified by CALIOP over the Baffin Sea and Greenland Sea as shown in Figure 14b and Figure 14c, respectively.

We observed a similar transport of biomass burning aerosol over the Mediterranean Sea, leading to a BER of 0.025 sr$^{-1}$ (Chazette et al., 2016) higher than the one retrieved here (~0.018±0.002 sr$^{-1}$). There is no reason for a typical BER value for biomass burning aerosols. Indeed, the BER is highly dependent on the chemical composition of aerosols via the complex refractive index, but also on their size distribution. Furthermore, both size distribution and chemical composition of biomass burning aerosols depend on the type of combustion and the intensity of the fire. Moreover, aerosols age during transport. Hence, a wide range of BER values is likely for biomass burning aerosol after a long-range transport (Amiridis et al., 2009).




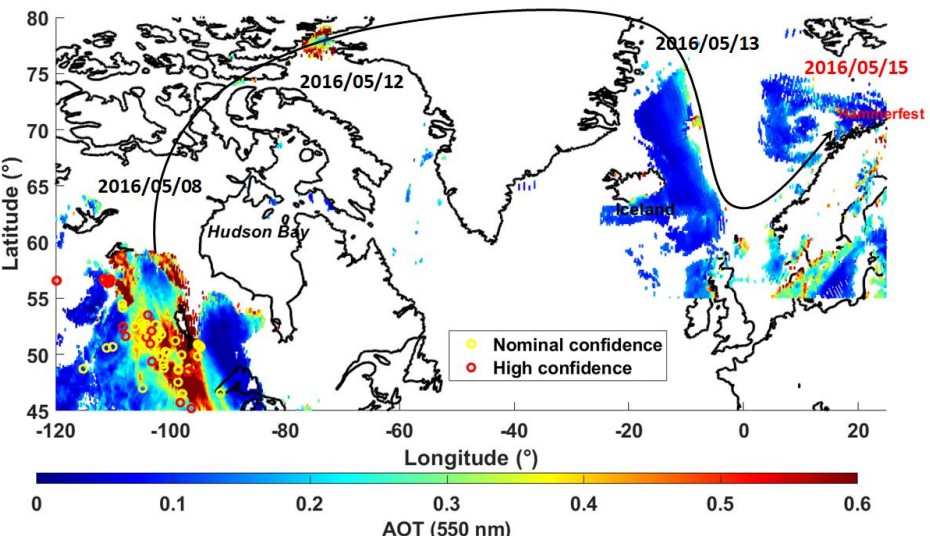


**Figure 13: MODIS-derived aerosol optical thickness (AOT) at 550 nm for three different days and locations. The dates**
**are indicated in the figure. The thermal anomalies derived from the MODIS fire product are also given on 8 May, 2016,**
**corresponding to the origin of the studied aerosol plume studied. The route followed by the biomass burning plume is**
**represented by a black solid line. It begins on 8 May, to finish on 15 May.**

a)

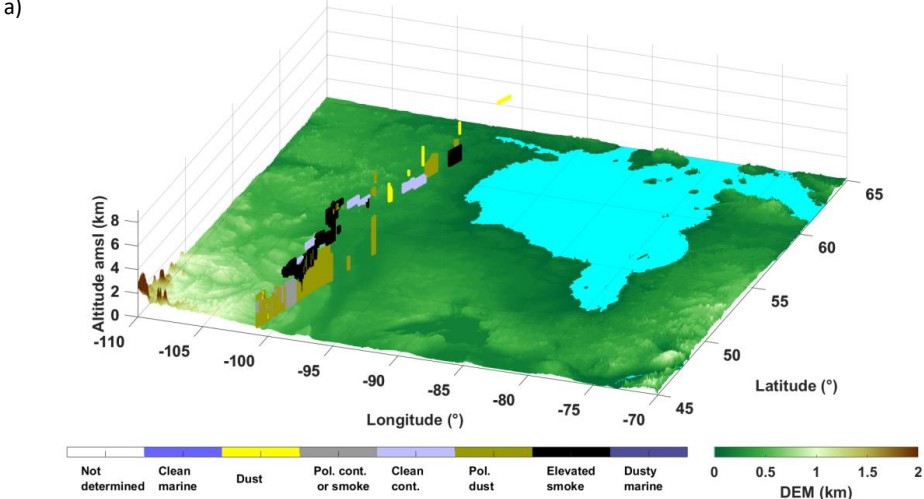



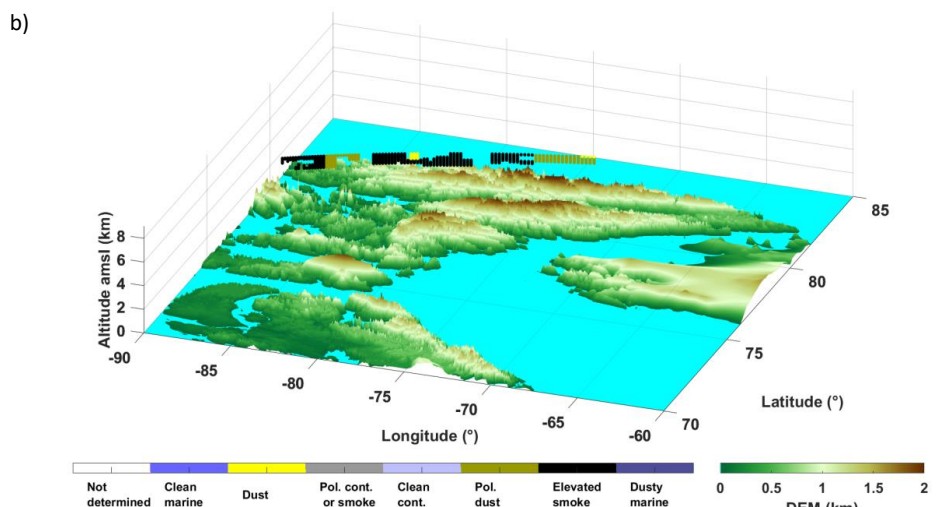

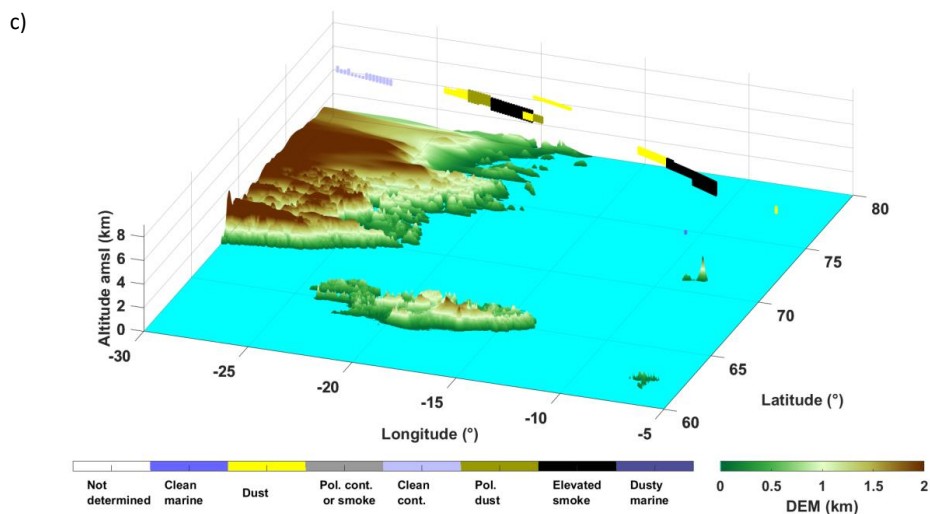

Figure 14: CALIOP-derived aerosol typing for a) the Fort McMurray on 8 May, b) the Baffin Sea on 12 May, and c) the Greenland Sea on 13 May, corresponding to the plume identified by MODIS in Figure 13.

## 5.2    Aerosol plume on 20-21 May

As for the previous aerosol plume, the origin seems to be from Canada. The back trajectories show potential contributions from Russia, but checking the spaceborne observations corresponding with the potential plume location, we do not identify any forest fires or anthropogenic emissions. The Canadian origin could not be clearly established from MODIS observation due to strong cloud cover. A large plume (AOT> 0.8) is found over the St. Lawrence region on 12 May, (Figure 15a) and corresponds to the transport of air masses along the back trajectories. Continuing the back trajectories, the Fort McMurray area, where forest fires have persisted, also appears to be the main source. An orbit of CALIPSO passes over the eastern part of the plume on 12 May, and



shows that it is mainly composed of elevated smoke aerosols from Canada (Figure 15b). The BER that has been
found (0.012 sr$^{-1}$) can also be attributed to biomass burning aerosols. However, given the possible values, it is not
a criterion.

a)

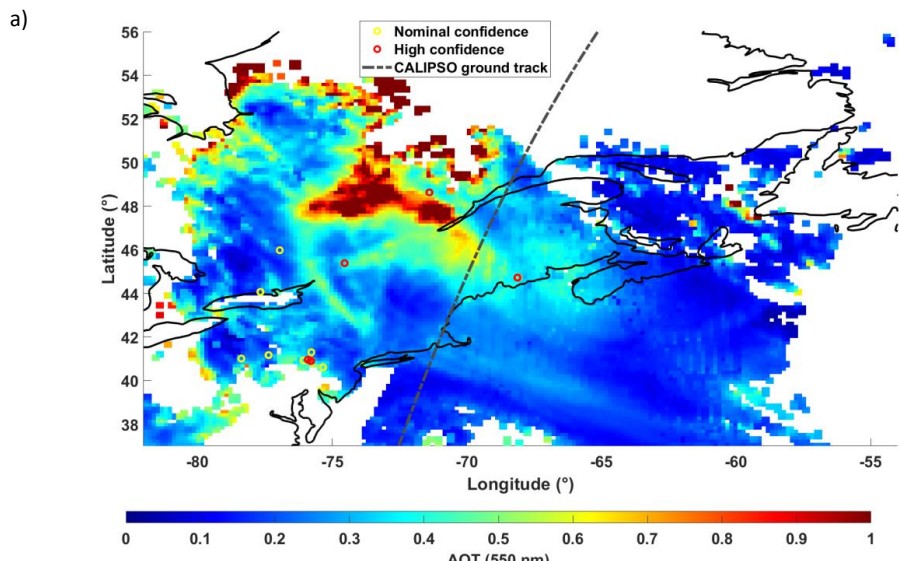

b)

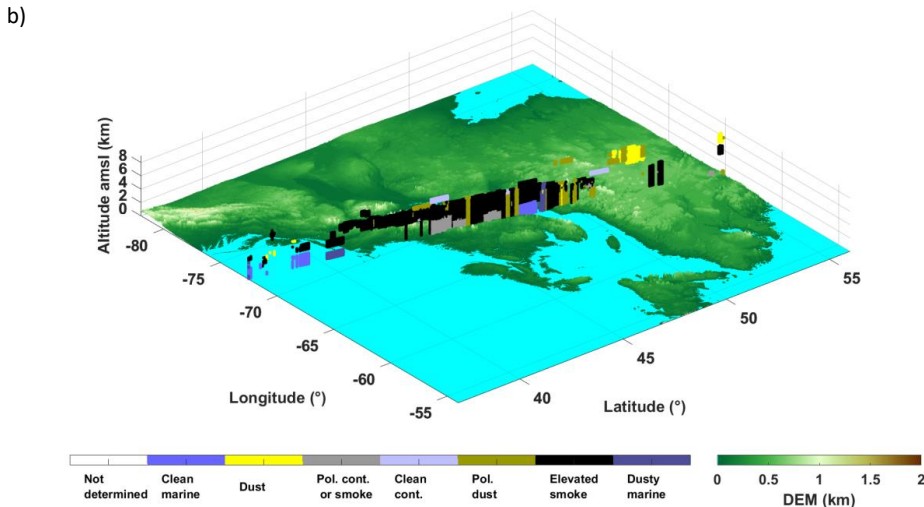


**Figure 15: a) MODIS-derived aerosol optical thickness (AOT) at 550 nm and thermal anomalies on 8 May, 2016; b)**
**CALIOP-derived aerosol typing (orbit 2016-05-12T06-53-10ZN). The CALIPSO ground track is indicated in a).**
**5.3     Aerosol plume on 22-23 May**

10.5194/acp-2018-426
Atmospheric Chemistry and Physics



The origin of this last aerosol plume is more easily identified to be the Canada, also in the area of Fort Mc Murray,
on 15 May. The aerosol plume emitted by the forest fires is well circumscribed by MODIS with AOTs greater
than 1. The locations of the fires are also indicated by the thermal anomaly. The CALIPSO orbit passes just above
the plume and offers the possibility to characterize the aerosols as elevated smoke, polluted continental or smoke
and polluted dust. As for the aerosol plume on 20-21 May, the same remark can be made on the derived BER of
0.013 sr⁻¹.

a)

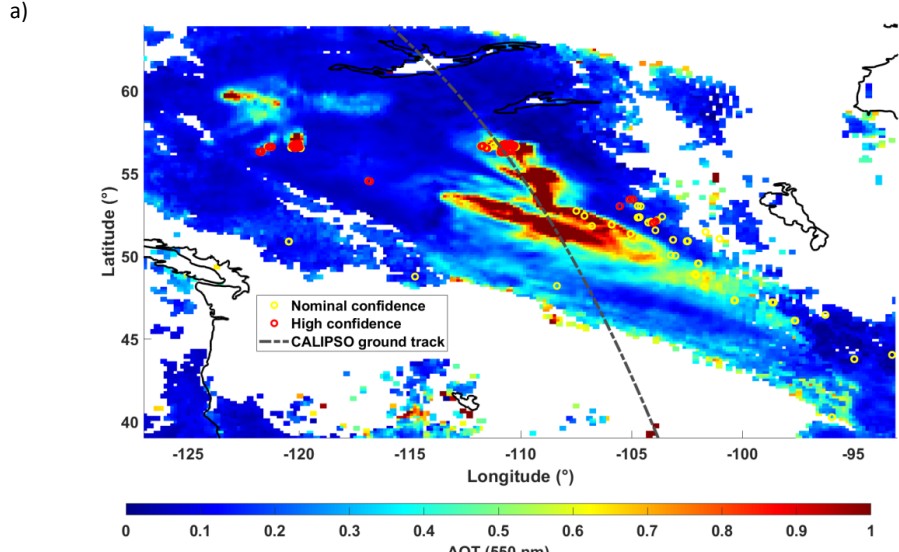

b)

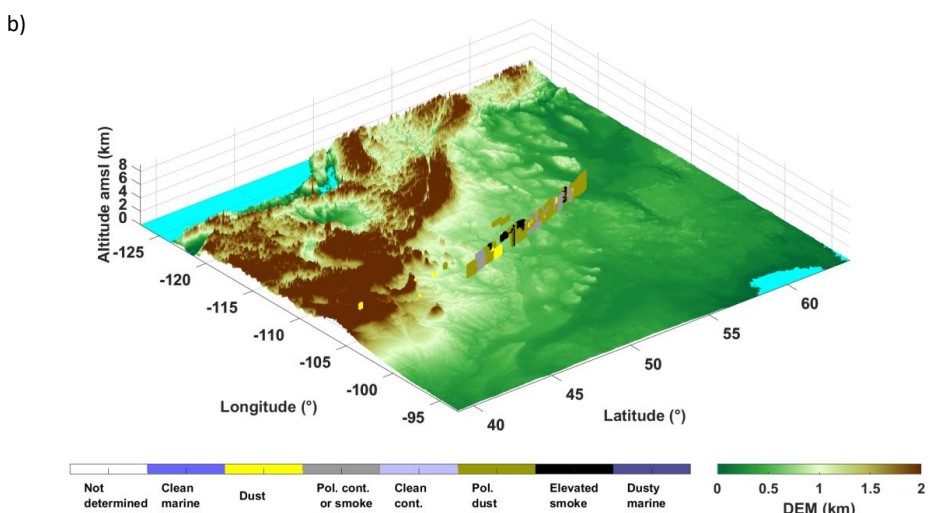




**Figure 16: a) MODIS-derived aerosol optical thickness (AOT) at 550 nm and thermal anomalies on 15 May, 2016; b)**
**CALIOP-derived aerosol typing (orbit 2016-05-15T19-42-56ZD). The CALIPSO ground track is indicated in a).**
**6 Data coherence**
**6.1 Coherence on the vertical profiles**
For higher altitude aerosol layers, we do not have any airborne observations to check the consistency of the results
with the lidar embedded on the ULA. Nonetheless, we have that possibility for the lower troposphere. Figure 17
shows the comparison between different approaches to retrieve the AEC vertical profile within the first 2 km of
the atmosphere. Horizontal and nadir lines of sight measurements performed from the ULA are compared for the
4 flights considered. We consider the closer 10 nadir profiles from the location of the spiral ascent (or descent). In
all the cases, the AEC profiles derived from the different approaches are all in agreement within 0.01 km$^{-1}$ of
uncertainty.
On 16 May, ground-based lidar data are not available due to low cloud cover. For the three other days, the 20
profiles closer in time to the airborne lidar profiles are considered. They are plotted with a solid line, together with
their error bars in Figure 17b-d. For the flights 10 and 11 a slight underestimation is noted, but error bars overlap
(within ~0.01 km$^{-1}$). The WALI-derived AEC profile is a better match with the ones derived from the airborne
lidar for flight 13, except in the PBL where they highlight a larger AEC. Such a discrepancy may be due to the
fact that measurements from the ULA were mainly preformed over the ocean (Figure 11a). Note that the AEC
profile derived from nadir measurement is not drawn with its rms to lighten the figure, knowing that it is like that
of other flights.

a)
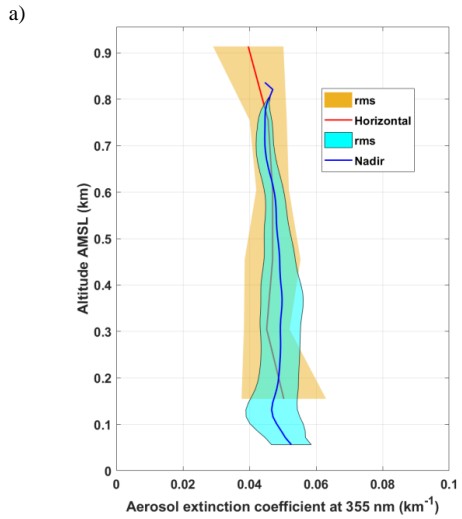

b)
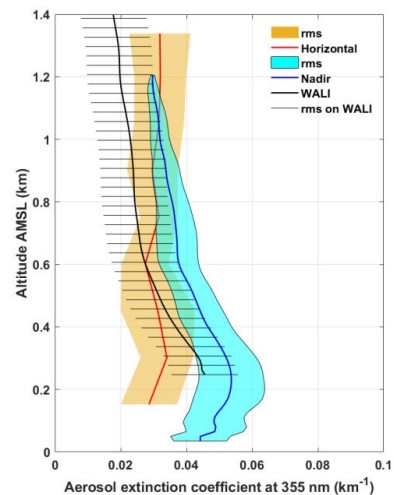





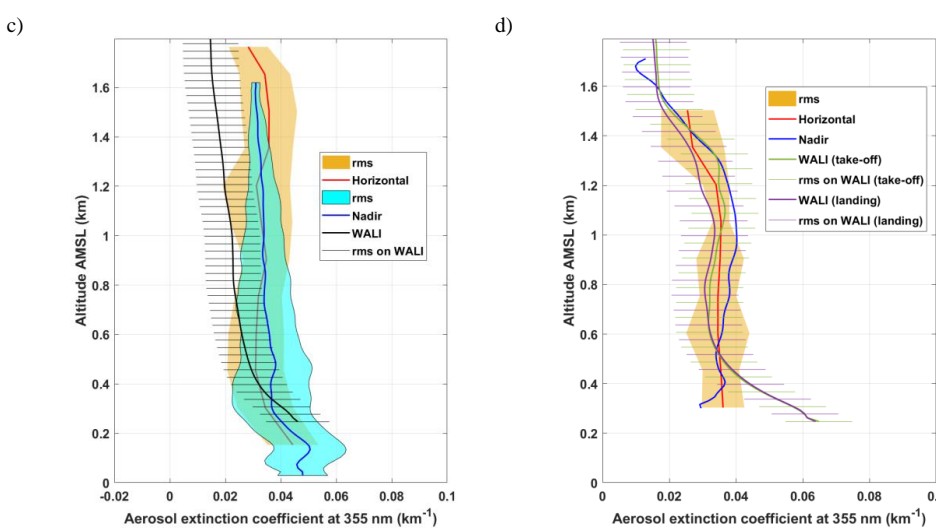

**Figure 17: Vertical profiles of the AEC derived from the airborne and ground-based lidars for times corresponding to a) flight 4, b) flight 10 & 11, and d) flight 13.**

## 6.2    Coherence on the aerosol optical thickness

Lidar-derived AOTs are checked against a SOLAR Light® Microtops II manual sunphotometer. The measurements were performed in clear sky condition during the three observation periods presented in Table 3. Measurements have not been continuous, since they have been carried out alternatively with lidar observations. On 13 and 14 May, mean AOT at 355 nm of $0.059 \pm 0.005$ is derived and matches very well the value retrieved from lidar measurement outside the upper aerosol plume. In the same conditions, we report AOTs of $0.084 \pm 0.005$ and $0.073 \pm 0.005$ on 19 and 20 May, respectively. Note that manual solar targeting induces an additional non-systematic bias, which leads to an absolute uncertainty assessed as of the order of 0.03 when comparing with simultaneous measurements by an automated sunphotometer before the field campaign.

We note a low background AOT over Hammerfest, which is between 0.06 and 0.08 at 355 nm (~$0.04\pm0.01$ at 550 nm). Such a value appears to match the one derived from the available MODIS data leading to ~$0.05\pm0.06$ during the entire field campaign. To consider a longer time frame, we give the histograms of AOT and Ångström exponent from 2008 to 2016 for the closer AERONET station of Andenes (69N 16E, ~320 km southwest of Hammerfest) in  Figure 18. The mean AOT at 355 nm is lower than 0.1 with a standard deviation of ~0.5. The Ångström exponent is very variable, mainly between 0.5 and 2, due to long-range transport aerosol (anthropogenic pollution, biomass burning and Saharan dust) originated in central and eastern Europe (Rodríguez et al., 2012). Note that the Ångström exponent derived from the manual sunphotometer is between 1.2 and 1.7, when considering the wavelengths of 380 and 500 nm.



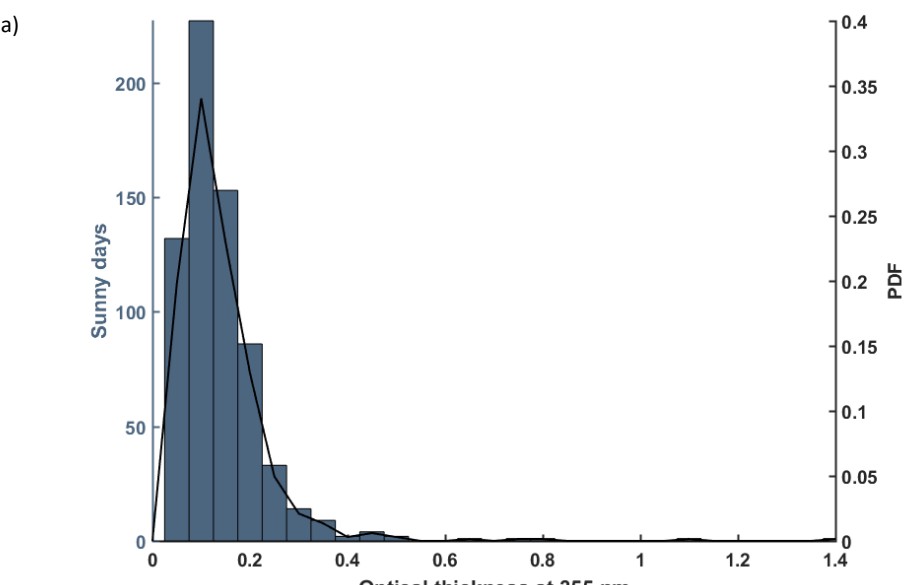

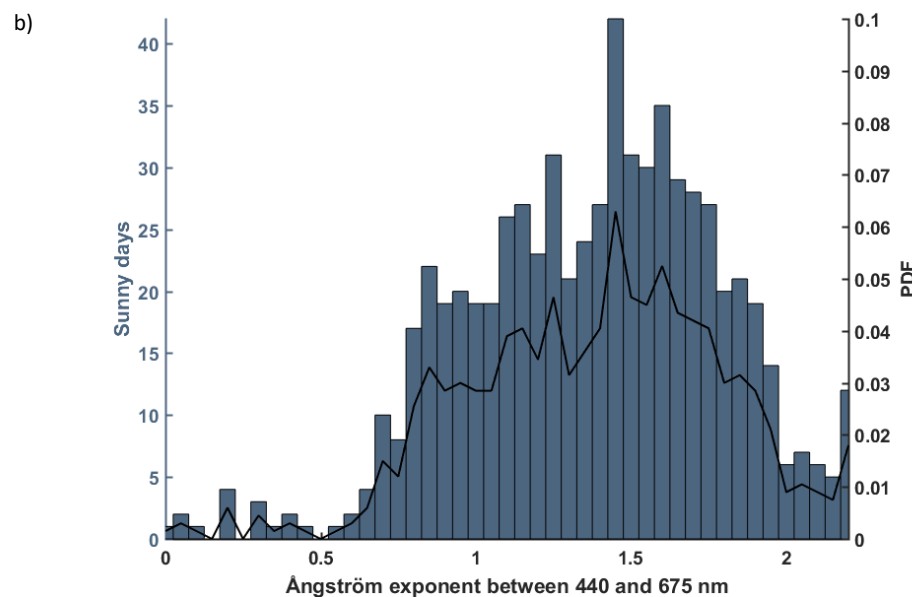

**Figure 18: Histograms of a) the aerosol optical thickness at 355 nm and b) the Ångström exponent between 440 and 675**
**nm for the AERONET station of Andenes (69N 16E). The data are for the clear days between 2008 and 2016. The**
**probability density functions (PDF) are also given.**
**7    Conclusion**



This work contributes to shed light on the abundance of aerosols in late spring over the European Arctic. During
the PARCS field-campaign, from 13 to 26 May, 2016, we collected an original dataset of remote sensing
measurements performed with ground-based and airborne (ULA) lidars. We evidenced 3 cases of aerosol long-
range transport over 2 weeks, originating from the Fort McMurray area, where strong forest fires occurred. They
followed different pathways to reach Northern Norway, but they significantly increased the AOT by a factor of up
to ~2. The AOT was enhanced from a background value of ~0.08 (~0.05), if not less, to ~0.2 (0.12) at the
wavelength of 355 nm (550 nm). This may imply a strong influence of long range transport of biomass burning
aerosols on the radiative budget over the Arctic area.
In the lower troposphere, below 3 km AMSL, the aerosol load is weak and corresponds to the previously observed
background value. In Hammerfest, airborne lidar measurements have shown a strong homogeneity of the PBL.
The main causes inducing a heterogeneity are i) the marine aerosol production, which is a function of the surface
wind speed, ii) the advection of northern air masses from industrial sites in Russia (Murmansk region), and iii) the
contribution of the Melkoya facility flares. We noted a very local effect of the active low-pressure flare, with an
enhancement close to 0.02 of the AOT at 355 nm. The effect on the environment therefore appears to be weak.
Because this plant is rather isolated, extending the conclusions to larger oil and gas rigs like those identified in
Figure 1 is hardly possible and would be purely speculative.
From an experimental perspective, the coupling between ground-based and airborne lidar measurements proved
to be essential for data analysis. The lidar systems are complementary and the coupled approach allows
confirmation of the results. With ULA flights, however, we remain in the vicinity of the ground station and flights
with larger carriers would be more suited to the regional scale. Nevertheless, one would lose in flexibility of
execution and in repetitiveness of measurement, inevitably limited by the cost of the flights.
**Acknowledgments:** This work was supported by the French Institut National de l'Univers (INSU) of the Centre
National de la Recherche Scientifique (CNRS) via the French Arctic initiative. The Commissariat à l'Energie
Atomique et aux énergies alternatives (CEA) is acknowledged for its support. We thank Yoann Chazette, Nathalie
Toussaint and Sébastien Blanchon for their help during the field experiment. The ULA flights were performed by
Franck Toussaint. The Avinor crew of Hammerfest Airport, represented by Hans-Petter Nergård, and the company
Air Creation company are acknowledged for their hospitality. Dr. K. S. Law is acknowledged for securing the
funding of the Pollution in the ARCtic System campaign. Support on computer modeling by T. Onishi was granted,
allowing access to the HPC resources of IDRIS under the allocation A001017141 made by GENCI, and to the
IPSL mesoscale computing center (CICLAD: Calcul Intensif pour le Climat, l'Atmosphère et la Dynamique).

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
