# Peer review of "Springtime aerosol load as observed from ground-based and airborne lidars over Northern Norway"

_Atmospheric Chemistry and Physics, 2018_

## Referee Comment (RC1) · Anonymous Referee #3 · 13 Jul 2018

The article is novel and interesting, it has a sufficient impact and adds to the knowledge base of the long-range transport of anthropogenic and biomass burning aerosols over the Arctic region. This work presents an original dataset of remote sensing measurements performed with ground-based and airborne lidars over the North of Norway (Hammerfest) during almost three weeks in May 2016. The ground based lidar measurements along with satellite and model data (FLEXPART-WRF) highlighted 3 interesting cases of long range transported aerosol over Norway, from strong forest fires that occurred in the area of Fort McMurray, in Canada. The authors specifically focus on the measurements both from ground and air borne then enhance the work by proving the coherence on the aerosol optical thickness, coherence on the vertical profiles

and investigating the origin of the aerosol layers. This topic is of high importance for both the lidar research community and satellite. The planning and implementation of the campaign is well done and a great example for other campaigns (e.g. for calibration/validation of satellites missions) The paper is written well and logically organized. The findings are supported by clear and well-presented analyses. Overall, I consider this manuscript easy to be read with important findings and therefore I recommend the publication.
* * *

---

## Referee Comment (RC2) · Anonymous Referee #2 · 25 Jul 2018

General

The paper is well written and presents interesting aerosol measurements with a small airborne lidar (aboard an ultra-light aircraft) and a mobile ground-based lidar (in a small van). The authors report aerosol measurements conducted in the European Arctic, a region for which aerosol profile observations are rare. The case studies show that lofted layers advected from polluted regions and wildfire places in North America can easily reach rmote and pristine areas. The paper also shows an interesting approach how to characterize individual pollution sources (flares of the Melkoya gas processing facility). In view of the intensive discussions on arctic climate change the paper comes

at the right time.

Minor revisions may improve the paper.

P4, L107 to P6, L133: Please provide some information (a small paragraph plus reference) on the calibration of 355 nm depolarization-ratio observations. In addition, the determination of the 355 nm particle depolarization ratio is not so easy compared to computation, e.g., at 532 nm. Please provide uncertainty information. At very low values of the particle backscatter coefficient, the determination of the particle depolarization ratio at 355 nm is no longer possible (error exceeds 100%). On the other hand, depolarization ratio observations contain information on the aerosol type and particle shape. So it should be know how good the observations are.

P9, L220 to P14, L259: The BER (or the inverse value. . .. the lidar ratio) is an important parameter to characterize the aerosol type. Please provide as often as possible (in the text and the figures) also the respective lidar ratio values. In 95% of all reports on BER or lidar ratio, the lidar ratio is used. So, the lidar community is familiar with lidar ratio values and not with BER values.

P13 to P14: Figure 8, I would like to see the BER uncertainties as numbers in the plots as well.

P14, L 286: Regarding the flare emissions. I speculate that the freshly produced particles are very small, and most are too small to be optically active already. They may become increasingly activate after growth and coagulation , . . ..downwind after several hours. . .. Please comment on that.To state that the AOT contribution is 0.02 or 0.07 may be missleading, the overall effect may be much higher. . . Please discuss this point a bit.

---

## Author Comment (AC1) · 3 Aug 2018

**Response to reviewers**

Dear Editor, please find hereafter the response to the referee's comments. We thank the reviewers for thoughtful and constructive comments on our manuscript. We appreciate the time s/he invested in the review. We believe that our revised manuscript addresses all the comments.

In the following, the comments made by the referees appear in black italic, while our replies are in bold, and the proposed modified text in the typescript is in blue.

**Reviewer #1**

We thank the reviewer for his positive remarks. The field-experiment exposed in this paper was mainly dedicated to the assessment of the aerosol load above the polar circle. The main reason of this project was indeed the lack of data for climate modeling.

**Reviewer #2**

Minor revisions may improve the paper.

*P4, L107 to P6, L133: Please provide some information (a small paragraph plus reference) on the calibration of 355 nm depolarization-ratio observations. In addition, the determination of the 355 nm particle depolarization ratio is not so easy compared to computation, e.g., at 532 nm. Please provide uncertainty information. At very low values of the particle backscatter coefficient, the determination of the particle depolarization ratio at 355 nm is no longer possible (error exceeds 100%). On the other hand, depolarization ratio observations contain information on the aerosol type and particle shape. So it should be know how good the observations are.*

**Yes, the PDR is a very important parameter for aerosol typing. We have already described the calibration process in Chazette et al. (2012) and we have added this reference in the text. For AEC values lower than 0.03 km$^{-1}$, the PDR is not computed for the exact reason explained by the reviewer. The reference to Dieudonné et al. (2017) where a complete uncertainty study was performed on the PDR derived by a similar system has been added.**

The aerosol extinction coefficient (AEC), the backscatter to extinction ratio (BER, inverse of the lidar ratio (LR)) are derived following Chazette et al. (2014) and references therein. The calibration process to retrieve the particle depolarization ratio (PDR) is given in Chazette et al. (2012). The absolute uncertainties on the AEC are ~0.01 km$^{-1}$ and the ones on the PDR are ~1-2% for AEC > 0.03 km-1. For smaller AEC, the error on the PDR is too high and we do not compute it. An example on different aerosol types is given in the Appendix A of Dieudonné et al. (2017).

*P9, L220 to P14, L259: The BER (or the inverse value. . .. the lidar ratio) is an important parameter to characterize the aerosol type. Please provide as often as possible (in the text and the figures) also the respective lidar ratio values. In 95% of all reports on BER or lidar*

*ratio, the lidar ratio is used. So, the lidar community is familiar with lidar ratio values and not with BER values.*

**Yes, the LR is often used in the literature and we have added some values in the text.**

P13 to P14: Figure 8, I would like to see the BER uncertainties as numbers in the plots as well.

**We have added the uncertainties in the legends.**

*P14, L 286: Regarding the flare emissions. I speculate that the freshly produced particles are very small, and most are too small to be optically active already. They may become increasingly activate after growth and coagulation , . . ..downwind after several hours. . .. Please comment on that. To state that the AOT contribution is 0.02 or 0.07 may be missleading, the overall effect may be much higher. . . Please discuss this point a bit.*

**It is difficult to respond to this interesting remark because additional means would have been necessary. We have added the following explanation:**

The aerosol particles may age in different ways. These processes depend on the initial chemical composition which will lead to the coagulation and/or the adsorption of gaseous molecules on the surface of the existing aerosols. In general, this process is quite fast and occurs when relaxing in the atmosphere, i.e. at the exit of the chimney. The particles thus formed may be more or less reactive, and more or less hygroscopic. Their size distribution, as well as their complex refractive index can change, especially in the presence of relative humidity greater than 50-60% (Randriamiarisoa et al., 2006). They can therefore become more scattering, and generally less absorbent. The AOT may therefore increase during their aging in the atmosphere. We cannot afford to propose more insight about this phenomenon because of the lack of in-situ chemical analysis during the field experiment.

---

## Author Comment (AC2) · 3 Aug 2018

See attached file

Please also note the supplement to this comment:
https://www.atmos-chem-phys-discuss.net/acp-2018-426/acp-2018-426-AC2-supplement.pdf

———————————————————